# Molecular Epidemiological Survey for Degenerative Myelopathy in German Shepherd Dogs in Japan: Allele Frequency and Clinical Progression Rate

**DOI:** 10.3390/ani12131647

**Published:** 2022-06-27

**Authors:** Shinichiro Maki, Md Shafiqul Islam, Tomohito Itoh, Masanobu Nurimoto, Akira Yabuki, Yu Furusawa, Hiroaki Kamishina, Yui Kobatake, Tofazzal Md Rakib, Martia Rani Tacharina, Osamu Yamato

**Affiliations:** 1Laboratory of Clinical Pathology, Joint Faculty of Veterinary Medicine, Kagoshima University, Kagoshima 890-0065, Japan; k6993382@kadai.jp (S.M.); si.mamun@ymail.com (M.S.I.); yabu@vet.kagoshima-u.ac.jp (A.Y.); rakibtofazzal367@gmail.com (T.M.R.); martia.rt@fkh.unair.ac.id (M.R.T.); 2Department of Pathology and Parasitology, Faculty of Veterinary Medicine, Chattogram Veterinary and Animal Sciences University, Khulshi, Chattogram 4225, Bangladesh; 3Maebashi Institute of Animal Science, Livestock Improvement Association of Japan, Inc., 316 Kanamaru, Maebashi 371-0121, Japan; titoh@liaj.or.jp (T.I.); nurimoto@liaj.or.jp (M.N.); 4Kagoshima University Veterinary Teaching Hospital, Joint Faculty of Veterinary Medicine, Kagoshima University, Kagoshima 890-0065, Japan; k9110942@kadai.jp; 5Joint Department of Veterinary Medicine, Faculty of Applied Biological Sciences, Gifu University, Gifu 501-1193, Japan; kamishinah@gmail.com (H.K.); kobatake@gifu-u.ac.jp (Y.K.); 6KyotoAR Veterinary Neurology Center, 208-4 Tai-Shiarai, Kumiyama-cho, Kuze-gun, Kyoto 613-0036, Japan; 7Faculty of Veterinary Medicine, Airlangga University, Mulyorejo, Surabaya 60115, Indonesia

**Keywords:** canine *SOD1* gene, degenerative myelopathy, dog breeding, German Shepherd Dog, mutant allele frequency, disease prevention

## Abstract

**Simple Summary:**

Canine degenerative myelopathy (DM) is an adult-onset, chronic, progressive neurodegenerative disease caused by a DM-associated mutation (*SOD1*:c.118G>A, p.E40K) that commonly occurs in German Shepherd Dogs (GSDs). This study aimed to determine the mutant allele frequency in the Japanese GSD population and to analyze the clinical progression rate among GSDs with the homozygous mutant A/A genotype. The survey found 330 G/G dogs (61%), 184 G/A dogs (34%), and 27 A/A dogs (5%) among the 541 dogs examined, indicating that the mutant allele frequency was 0.220; the Japanese GSD population can be considered to be in Hardy–Weinberg equilibrium. Clinical analysis revealed that the clinical progression rate was particularly high (100%) among A/A dogs aged >10 years. Appropriate mating management is crucial for the management and prevention of DM in the Japanese GSD population.

**Abstract:**

Canine degenerative myelopathy (DM) is an adult-onset, chronic, progressive neurodegenerative disease reported in multiple canine breeds, including the German Shepherd Dog (GSD). Clinical signs include progressive motor neuron paralysis, which begins in the pelvic limbs and eventually leads to respiratory distress, which may necessitate euthanasia. A common DM-associated mutation is a single nucleotide substitution that causes an amino acid substitution (c.118G>A, p.E40K) in the canine *SOD1* gene. This *SOD1* mutation and the clinical progression rate of A/A risk genotype in the Japanese GSD population have not been analyzed before. Therefore, the aim of this study was to determine the frequency of the mutated allele and analyze the clinical progression rate in the Japanese GSD population. We studied 541 GSDs registered with the Japanese German Shepherd Dog Registration Society between 2000 and 2019. Genotyping was performed using real-time PCR with DNA extracted from the hair roots of each dog. The study revealed 330 G/G dogs (61%), 184 G/A dogs (34%), and 27 A/A dogs (5%), indicating a frequency of the mutant allele of 0.220, which are in Hardy–Weinberg equilibrium. We analyzed the clinical signs in A/A dogs with an age limit of 10 years based on information obtained from the dogs’ owners. Of the seven A/A dogs older than 10 years, owners reported DM-related clinical signs, indicating a clinical progression rate of 100%. These results, further genotyping, and thorough clinical examinations of *SOD1* A/A risk genotype will help control and prevent DM in the Japanese GSD population.

## 1. Introduction

Canine degenerative myelopathy (DM) is an adult-onset, chronic, progressive neurodegenerative disease that occurs in multiple canine breeds (Online Mendelian Inheritance in Animals (OMIA) 000263-9615), including the Pembroke Welsh Corgi (PWC), German Shepherd Dog (GSD), Boxer, Collie, and Bernese Mountain Dog (BMD) [1,2,3,4]. DM can be used as an animal model for human amyotrophic lateral sclerosis (ALS) (Online Mendelian Inheritance in Man (OMIM) #105400 ALS1) [5]. DM begins to affect dogs at 8 years of age, especially affecting those over 10 years of age at the onset of clinical signs, which include progressive, asymmetric paraparesis of the pelvic limbs that progresses to generalized proprioceptive ataxia, and a lack of paraspinal hyperesthesia [1,2,3,6,7]. These clinical signs ultimately lead to paraplegia and dyspnea, which necessitate euthanasia. Such disease progression is persistent, and dog owners often elect euthanasia within a few years of the onset of clinical signs, especially when their dog becomes paraplegic [2,3]. Only symptomatic treatments such as oxygen inhalation and oral supplement containing curcumin are available in dogs affected with DM [7].

In the spinal cord of DM-affected dogs, cytoplasmic accumulation of the aggregate form of the Cu/Zn superoxide dismutase 1 (SOD1) protein is observed in the neuronal cells, which can be detected using anti-SOD1 antibodies; this is a characteristic sign closely linked with DM pathogenesis, resulting in neuronal cell death and spinal degeneration [4,8,9,10,11]. The most common DM-associated mutation is a single nucleotide substitution that causes an amino acid substitution (c.118G>A, p.E40K) in the canine *SOD1* gene, and the mutation has been found in 124 different canine breeds [4,8]. Among these, DM is particularly common in a few breeds, such as the PWC, GSD, Boxer, Collie, and BMD, likely due to high mutant allele frequencies within these breed populations (0.37–0.79) [4]. Another DM-associated missense mutation (c.52A>T, p.S18T) is only found in BMDs [12]; therefore, two molecular tests for c.118G>A and c.52A>T mutations are necessary for DM diagnosis in BMDs.

Clinical signs and progression are relatively uniform among dogs of the same breed and between different breeds [3,8]. This uniformity could be explained by the fact that DM in dogs except for BMDs is associated with the same c.118G>A mutation [8]. Some BMDs have c.52A>T mutation that is also associated with DM [12]. The c.118G>A mutation is responsible for a p.E40K amino acid substitution, which is likely to cause the formation of misfolded proteins, and eventually aggregates and accumulates in the cytoplasm of neurons [8,9,13]. Although the whole mechanisms underlying mutant SOD1 toxicity have remained unclear, there is some evidence in DM and ALS that misfolded SOD1 inhibits axonal transport [14,15] and impairs organelle functions resulting in endoplasmic reticulum stress [16] and mitochondrial dysfunction [17,18].

Various factors may accelerate or delay disease progression, such as the canine *SP110* gene found as a modifier in PWCs [19]. Due to this complex background, clinical diagnosis based on typical clinical manifestations and risk genotypes of the *SOD1* gene (c.118A/A, c52T/T, and a compound heterozygote of the two mutations) through molecular testing is difficult, although a definitive diagnosis can be made based on histopathological examination that shows spinal degeneration in combination with immunohistochemical analysis of cytoplasmic aggregated SOD1 protein [2,3,9,10,20].

To the best of our knowledge, molecular epidemiological surveys for DM have yet to be conducted in the GSD in Japan, although a survey was performed in PWCs [21] and Collies [10]. This study aimed to determine the mutant allele frequency for risk assessment of DM and to analyze the clinical progression rate in the Japanese GSD population registered with the Japanese German Shepherd Dog Registration Society (JSV) [22], which is affiliated with the Weltunion der Vereine für Deutsche Schäferhunde (WUSV), the world union of GSD associations [23].

## 2. Materials and Methods

This study was carried out in accordance with the guidelines regulating animal use and ethics at Kagoshima University (no. VM15041; approval date: 29 September 2015). Informed oral consent was obtained from the participating owners through the JSV.

### 2.1. Sample Collection and Genotyping

Samples were collected from 541 JSV-registered GSDs born between 6 January 2000 and 31 March 2019. DNA was extracted from fur samples using a commercial kit (QIAamp DNA Micro Kit, Qiagen, Hilden, Germany) according to the manufacturer’s protocol. In the ten fur samples that failed for genotyping using this kit, DNA was extracted using a different commercial kit special for DNA extraction from hair samples (ISOHAIR, Nippon Gene, Tokyo, Japan) according to the manufacturer’s protocol. The genotypes of DM-associated mutation (*SOD1*:c.118G>A), that is, wild-type homozygote (G/G), heterozygous carrier (G/A), and mutant homozygote (A/A) were determined using a real-time PCR assay, as previously reported [21].

### 2.2. Statistical Analysis

The allele frequencies obtained in this and previous studies [4,10,24,25,26] were analyzed using the chi-squared test for Hardy–Weinberg equilibrium. Deviations between the measured and expected values were regarded as statistically significant at *p* < 0.05, indicating that data at *p* ≥ 0.05 is in Hardy–Weinberg equilibrium. The mutant allele frequency in the Japanese GSD population in this study was compared with that of the GSD populations in other countries [4,10,24,25,26] and in Japanese PWCs and Collies [10,21]. Differences were analyzed using Fisher’s exact test, with *p* < 0.05 considered to be a statistically significant difference. Statistical analyses were performed using R software.

### 2.3. Clinical Progression Rate

The clinical progression rate was calculated based on the number of dogs with progressive DM-related clinical signs in the number of dogs with the risk genotype (A/A) determined by this study. The clinical progression rate was calculated separately in two groups of dogs aged under and over 10 years old. DM-related clinical signs were used to estimate the progression of clinical onset by a veterinarian (O.Y.) based on information obtained from owners of dogs with the risk genotype, as identified by a JSV GSD breeder (Appendix A).

## 3. Results

### 3.1. Allele Frequency

Genotyping of the Japanese GSD population revealed that among the 541 dogs surveyed, there were 330 G/G dogs (61.0%), 188 G/A dogs (34.0%), and 27 A/A dogs (5.0%) (Table 1 and Table 2). Based on this, we calculated a mutant allele frequency of 0.220, indicating expected frequencies of 0.60846, 0.34316, and 0.04838 for the G/G, G/A, and A/A genotypes, respectively. Chi-squared test analysis (χ^2^ = 0.042809, df = 2, *p* = 0.9788) indicates that these three genotypes were in Hardy–Weinberg equilibrium.

The data from previous studies on GSD populations in other countries (UK, USA, Poland, Israel, and Brazil) [4,23,24,25] were analyzed in the same way, resulting in the observed data in Hardy–Weinberg equilibrium, except for those in the UK [24] and the USA [4] in 2014 (Table 1). The mutant allele frequency in the Japanese GSD population (0.220) was significantly lower than the UK (0.377, *p* < 0.001) and USA (0.366, *p* < 0.001 in 2014 and 0.331, *p* < 0.01 in 2017), comparable to those in Poland (0.181) and Israel (0.175), and significantly higher than Brazil (0.121, *p* < 0.005), based on Fisher’s exact test.

Furthermore, the mutant allele frequency in the Japanese GSD population in this study was compared with the data previously reported in PWCs and Collies in Japan [10,21], which are also in Hardy–Weinberg equilibrium (Table 2). The mutant allele frequency in the GSD population (0.220) was significantly lower than that of the PWC population (0.697, *p* < 0.001). The mutant allele frequency in GSDs (0.220) was slightly higher than that in Collies (0.138), but there was not a significant difference between the two breeds.

### 3.2. Clinical Progression Rate

The owners of 14 out of 27 GSDs with the A/A risk genotype were interviewed to obtain information about their clinical signs. As a result, among the 27 GSDs (13 males, 12 females, and 2 unknown sex) aged between 5 years 4 months and 15 years 11 months, 9 GSDs (64.3%) were estimated to have developed DM-related clinical signs (Table 3). We further divided the dogs into two groups, older (7 dogs) and younger (5 dogs) than 10 years of age because the majority of DM-affected dogs are over 10 years old at the onset of clinical signs. Among the 5 GSDs younger than 10 years old, none had developed DM-related clinical signs, whereas all 7 GSDs over 10 years old had developed signs. This indicates that the clinical progression rate of DM in GSDs >10 years old recruited in this study was 100%. Two A/A dogs of unknown age had developed DM-related clinical signs.

## 4. Discussion

This study demonstrates that the mutant allele frequency of DM in the study population of GSDs was 0.220, and this mutation was distributed throughout the population according to the Hardy–Weinberg equilibrium. The mutant allele frequency in the Japanese GSD population was significantly lower than in the UK (0.377) [24] and USA (0.366 in 2014 and 0.331 in 2017) [4,26] (Table 1). However, the genotypes in the UK [24] and USA [4] in 2014 were not in Hardy–Weinberg equilibrium, suggesting that the samples may not have been collected entirely at random and were therefore more likely to include dogs carrying the mutation, for example, A/A dogs exhibiting DM-related clinical signs for an auxiliary diagnosis of DM. Therefore, the actual values in these two countries may be lower than these observed data, but other data collected in the USA (0.331) in 2017 that was in Hardy–Weinberg equilibrium had such high-risk genotype frequency that it indicates the USA likely has a higher mutant allele frequency than the countries surveyed. Furthermore, the values of the Japanese GSD population were comparable to those in Poland (0.181) and Israel (0.175) [26], and significantly higher than that in Brazil (0.121) [25]. The differences in the mutant allele frequencies between countries may be due to population stratification formed by the restrictions on the import and export of dogs. Therefore, prevention strategies for inherited diseases, such as DM, should be determined based on surveillance data for each breed in each country.

In Japan, molecular epidemiological surveys have been carried out for several inherited canine diseases to determine the associated mutant allele frequencies and thereby evaluate the necessity of prevention measures [27,28,29,30,31]. Among these, lethal diseases characterized by progressive neurological dysfunction include GM1 gangliosidosis in Shiba Inus (mutant allele frequency = 0.00509) [27] and Mame Shibas (0.00246) [28], neuronal ceroid lipofuscinosis (NCL) in Chihuahuas (0.00645) [30], Sandhoff disease in Toy Poodles (0.00101) [29], and NCL in Border Collies (0.0405) [31]. Compared with the mutant allele frequencies in these lethal diseases with juvenile, late juvenile, or early adult onset [32], DM had particularly high mutant allele frequencies in PWCs (0.697) and Collies (0.138) [10,21] as well as GSDs (0.220) in this study, although there was a notably large difference in frequency among these three breeds (Table 2).

The particularly high frequency of DM seems to be attributable to the fact that DM is an adult-onset, chronic disease with an incomplete recessive mode reported in the PWC, which was derived from the observation that not all PWCs with the A/A risk genotype develop DM-related clinical signs by age 15 [19]. As mentioned above, a haplotype (a combination of five single nucleotide polymorphisms) of the canine *SP110* gene that encodes the SP110 nuclear body protein involved in the modulation of gene transcription is found in PWCs with the risk genotype and is associated with an increased probability of developing DM as well as earlier onset of disease [19]. According to this complex clinical and molecular background, prevention measures should not be conducted based only on c.118G>A, especially in the PWC breed, which has a particularly high mutant allele frequency of 0.697 in Japan [21] and 0.792 in the USA [4]. Such prevention measures may risk unnecessarily increasing the inbreeding coefficient of the PWC population and thus the risks of other genetic diseases.

Unlike the clinical progression rate in PWCs, in the present study, we estimated the clinical progression rate in GSDs to be particularly high (100%) because among the examined GSDs aged over 10 years old, each dog developed DM-related clinical signs (Table 3). However, the rate might include misdiagnosis of other neurological diseases that cause similar motor disorders because the rate is based on clinical signs obtained from the owners during the interviews. Although the cause of this high clinical progression rate has yet to be confirmed, there may be a high incidence of the risk haplotype of the canine *SP110* gene in GSDs. Indeed, there are few Boxers with the A/A risk genotype that reached old age (>11 years old) without developing DM-related signs [19], suggesting that the clinical progression rate of Boxers is as high as that of GSDs, unlike PWCs. The high clinical progression rate in Boxers seems to be attributable to the high incidence of the PWC risk haplotype of the *SP110* gene [19]. Based on the mutant allele frequency (0.220), which is relatively low among the breeds predisposed to DM and the high clinical progression rate (100%), prevention measures could be conducted based only on c.118G>A in the GSD, unlike the PWC.

GSDs are the preferred choice of law enforcement and/or military agencies worldwide [33,34]. Due to extensive training, cost, and their valuable roles, the GSD is maintained in such a way as to provide the high quality of service for the longest possible period. Therefore, inherited diseases, such as DM, are one of the most serious problems in GSDs, which can result in various losses [34,35]. According to data (Appendix A) from the Japan Kennel Club (JKC) collected over years (1999–2021) [36], certified by the Federation Cynologique Internationale [37], an average of 483 (179–865) GSDs are registered annually by the JKC, which constitutes only 0.11% of the total dogs registered (an average of 406,353 per year). Japanese GSDs are mainly registered by the JSV [22], and the average number of GSDs registered per year is 556 (301–919), according to JSV records (Appendix A) for 11 years (2011–2021) (provided by the JSV). Some GSDs are registered as police dogs by the Nippon Police Dog Association [38], but these records are not publicly available. Based on the modes of GSD registration in Japan, the JSV approach used in this study is arguably the most effective way to conduct nationwide monitoring for inherited diseases in the Japanese GSD population. It is desirable that JSV publishes registered GSDs carrying mutations in preventable inherited diseases, including DM, and that breeders use the information to carry out appropriate breeding without producing the diseases.

## 5. Conclusions

This study demonstrates that the mutant allele frequency is notably high (0.220) in the Japanese GSD population, which is in Hardy–Weinberg equilibrium, and that the clinical progression rate is particularly high (100%) in GSDs with the A/A risk genotype over 10 years old. Therefore, the survey results and further genotyping of the DM-associated mutation (*SOD1*:c.118G>A) make a valuable contribution to the prevention of DM in the Japanese GSD population.

## Figures and Tables

**Table 1 animals-12-01647-t001:** Comparison of the mutant allele frequencies of degenerative myelopathy-associated mutation (*SOD1*:c.118G>A) in German Shepherd Dogs (GSDs) in Japan and elsewhere.

Country	Year Reported	Number of GSDs Examined	Number of GSDs with Each Genotype (%)			Mutant Allele Frequency	Statistics	
			G/G	G/A	A/A		Chi-Squared Test for Hardy–Weinberg Equilibrium	Fisher’s Exact Test (Difference vs. Japan)
Japan	2022 (this study)	541	330 (61.0)	184 (34.0)	27 (5.0)	0.220	*p* = 0.979 *	ND
UK	2014 [24]	150	66 (44.0)	55 (36.7)	29 (19.3)	0.377	*p* = 0.0273	*p* < 0.001 **
USA	2014 [4]	6458	3155 (48.9)	1884 (29.2)	1419 (22.0)	0.366	*p* < 0.001	*p* < 0.001 **
USA	2017 [26]	83	38 (45.8)	35 (42.2)	10 (12.0)	0.331	*p* = 0.908 *	*p* = 0.00722 **
Poland	2017 [26]	47	32 (68.1)	13 (27.7)	2 (4.3)	0.181	*p* = 0.901 *	*p* = 0.647
Israel	2017 [26]	20	14 (70.0)	5 (25.0)	1 (5.0)	0.175	*p* = 0.835 *	*p* = 0.612
Brazil	2020 [25]	95	72 (75.8)	23 (24.2)	0 (0.0)	0.121	*p* = 0.406 *	*p* = 0.00356 **

* Chi-squared test analysis (*p* ≥ 0.05) indicates that the observed data are in Hardy–Weinberg equilibrium. ** Fisher’s exact test analysis (*p* < 0.05) indicates that there is a significant difference versus the mutant allele frequency in the Japanese GSD population. ND: not determined.

**Table 2 animals-12-01647-t002:** Comparison of the mutant allele frequencies of the degenerative myelopathy-associated mutation (*SOD1*:c.118G>A) in Japanese canine populations of the German Shepherd Dogs (GSDs), Pembroke Welsh Corgis (PWCs), and Collies.

Breed	Year Reported	Number of Dogs Examined	Number of Dogs with Each Genotype (%)			Mutant Allele Frequency	Statistics	
			G/G	G/A	A/A		Chi-Squared Test for Hardy–Weinberg Equilibrium	Fisher’s Exact Test (Difference vs. GSD)
GSD	2022 (this study)	541	330 (61.0)	184 (34.0)	27 (5.0)	0.220	*p* = 0.979 *	ND
PWC	2013 [21]	122	11 (9.0)	52 (42.6)	59 (48.4)	0.697	*p* = 0.996 *	*p* < 0.001 **
Collie	2017 [10]	29	21 (72.4)	8 (27.6)	0 (0.0)	0.138	*p* = 0.670 *	*p* = 0.438

* Chi-square test analysis (*p* ≥ 0.05) indicates that the observed data are in Hardy–Weinberg equilibrium. ** Fisher’s exact test analysis (*p* < 0.05) indicates that there is a significant difference versus the mutant allele frequency in the Japanese GSD population. ND: not determined.

**Table 3 animals-12-01647-t003:** Clinical progression rate of degenerative myelopathy (DM) in German Shepherd Dogs (GSDs) with the risk genotype (*SOD1*:c.118A/A).

Age	Number of GSDs Examined *	Age Range	GSDs with DM-Related Clinical Signs **	Rate (%)
All	14	5 y 4 m–15 y 11 m	9	64.3
≥10 y	7	10 y 2 m–15 y 11 m	7	100
<10 y	5	5 y 4 m–9 y 5 m	0	0
Unknown	2	Undetermined	2	100

* Among the 27 GSDs with the risk genotype (A/A) found in this study, the owners of 14 GSDs were interviewed by a GSD breeder. ** DM-related clinical signs were estimated as the progression of clinical onset by a veterinarian (O.Y.) based on the information obtained from the owners through the above interviews.

## Data Availability

Not applicable.

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
