# Peer review of "Molecular Epidemiological Survey for Degenerative Myelopathy in German Shepherd Dogs in Japan: Allele Frequency and Clinical Progression Rate"

_animals, 2022, doi:10.3390/ani12131647_

Round 1

Reviewer 1 Report

Review of the paper titled “Molecular epidemiological survey for degenerative myelopathy in German Shepherd Dogs in Japan: allele frequency and clinical progression rate”

General comments

Aim of the reviewed manuscript was to investigate the allele frequency of a known mutation (SOD1: c.118G>A) associated with canine degenerative myelopathy in Japanese German Shepherd Dogs (GSD). Furthermore, the study proposed to analyse the clinical progression rate of the disease among dogs with a homozygous-mutant genotype.

The authors presented a straightforward and scientific-sound work. However, Materials and Methods lack some necessary information and, furthermore, Results and Discussion sections should be revised since there are a few points that needs to be addressed (see “specific comments by sections” below).

English language and style are fine, only a few details should be revised (please consider checking the use of semicolons).

In conclusion, I believe this paper should be reconsidered after MAJOR REVISIONS.

Specific comments by sections

Abstract

L44 – Could you rephrase the “which indicates HW equilibrium”? As it is, it looks like that the equilibrium is due to the frequency of 0.220.

L44-46 – This sentence takes for granted some information that is absent in the abstract and can only be found in the manuscript body (i.e., only A/A animals were analysed; age threshold of 10 years). Is it possible to add some information without excessing abstract’s length limitations?

Introduction

L55 – Why did you decide to not use abbreviations for Boxer and Collie breeds? Even if those names are composed of a single word, it would be nicer to have all the abbreviations in a similar format (i.e., 3 digits).

L81 – It is not clear what effect does SP110 gene have on disease progression… does it accelerate or delay the progression? Please clarify.

Material and Methods

L102 – please rephrase “DNA from the ten failed…” to something like “Ten fur samples failed DNA extraction with the first commercial kit. Therefore, a second commercial kit (info) was used”.

L113 – What is the rationale behind the comparison of allele frequency in GSD with the ones in Pembroke Welsh Corgi and Collies? Please explain in the manuscript. Furthermore, in the Introduction the authors mentioned the presence of this disease also in Boxer, and in Bernese Mountain Dog. Why allele frequency from these breeds were not included in the comparison in this study? Is it possible to further expand this study by adding these two breeds?

L117-119 – check “from”. I assume it is a ratio, but it is not clear from the authors’ sentence. Please rephrase.

L120-123 – info needed: which clinical signs were used for the clinical evaluation? How were the interviews performed? Is it possible to add the interview form as Supplementary Material?

Results

L132 – It would be useful to explicit each specific country as a list in brackets. This would help the authors to solve the issue presented in the next point.

L134 – The presence of two studies conducted in the USA is quite confounding. Please find a better way to differentiate the two studies other than the reference (e.g., USA-1 vs USA-2; use of the specific State where the study was performed…).

L158 – Probably a more formal sentence should be used here (e.g., “The owners of 14 out of 27 GSD with the A/A genotype were interviewed to obtain…”. What are the reasons for failed contacts with the remaining 13 A/A GSD? Please add this info in the manuscript.

L161-163 – Add in the manuscript the numbers of GSD composing the two groups. Further, explain the two “unknown” age animals from Table 3.

L163 – Add reference for the sentence “because the majority of DM-affected dogs […] signs”.

L166 – “[…] the clinical progression rate of DM in GSDs > 10 years old RECRUITED IN THIS STUDY was 100%”. The sentence as it is may lead to misinterpretations.

Discussion

L174 – How representative is the sample used by the authors (541 GSD) over the total Japanese GSD population? What is the size of the Japanese GSD population? Can you show the (approximate) percentage?

L206 – Which haplotype? How many mutations are involved in it?

L211 and L226 – please correct “c.118A” -> “c.118G>A”

L218-219 – The paper would largely benefit from adding this analysis to it. Since the “risk haplotype” is known (please add this information in the manuscript) why not testing it in the sampled GSD population?

Reviewer 2 Report

The manuscript describes the allele associated to degenerative myelopathy distribution in Japanese GSD population and present an analysis of clinical progression rate among homozygous mutant individuals.

In general, the manuscript presents interesting results that can contribute to deepen the knowledge aboute the disease in this particular population and also contribute to the overall understanding about the disease progression.

Minor considerations are detailed:

ABSTRACT
No coments.
INTRODUCTION
No coments.
MATERIAL AND METHODS
L98-99. It should be interesting if the authors could provide more information related to the animals. A supplementary table could be provided indicating Sex, Age and Genotypes for each individual used in the study.
L103-106. It be interesting if the autohrs could describe reaction conditions (primers and probe concentrations, final volume, mastermix concentrations and also cycling conditions), once the cited reference also cite other reference.
L117-123. It should be interesting if the authors could provide a more detailed information related to the clinical progression evaluation. Maybe a supplementary table including the questions (clinical signs) done during owner contact.

RESULT
No coments.
DISCUSSION
Authors sometimes simply compare the results obtained in this study with other published studies in other countries or other breeds.
CONCLUSIONS
No coments.

Reviewer 3 Report

The manuscript deals with canine degenerative myelopathy (DM) and its associated mutaion within the SOD1 gene (c.118G>A, 36 p.E40K) in the Japanese German Shepherd Dog (GSD) population. Allele frequencies indicate HWE. Among the 27 A/A dogs, 7 dogs reached an age > 10 years. These 7 A/A dogs appeared affected by DM.

Introduction

The survey on previous reports is rather incomplete or authors refer to previous reports later in the results. Some references are missing. The authors should provide one or two supplementary Tables summarizing currrent knowledge on SOD1 and DM (epidemiological studies and immunohistochemical and immunological studies).

Clinical progression: this is a critical issue because in GSD other diseases with similar signs may mimic DM. How did authors ensure that the DM diagnosis was validated through intensive examinations (neurological exams, MRT, CT). Can you rule out such as cauda equina syndrome or lumbosacral transitional vertebrae. Please extend your manuscript in this way and discuss these differential diagnosis, even add a supplementary Table to provide information on diseases causing problems with the hind legs in GSD.

Line 99-105: can you give some information how representative the samples collected are for the Japanese GSD population. So, it is necessary to give data on numbers of GSD puppies registered each year in Japan and size of breeding population per year. More information should be provided on study design and sampling strategy. JSV may provide pedigree data that you can look on the allelic distribution in breeding lines or clustering of the SOD1 A allele by coancestry.  At least some data and discussion on representativity of data have to be supplemented.

This issue is taken up at the end of discussion. But here, it is still hard to understand how representative the data are.

Line 143: rephrase sentence.

Line 165-166: clinical progression rate: please define this term. It is not clear to which disease you are referring.

The manuscript has to be better organized as some important information is hidden in the results and discussion section.

Round 2

Reviewer 1 Report

Review R1 of the paper titled “Molecular epidemiological survey for degenerative myelopathy in German Shepherd Dogs in Japan: allele frequency and clinical progression rate”

General comments

Aim of the reviewed manuscript was to investigate the allele frequency of a known mutation (SOD1: c.118G>A) associated with canine degenerative myelopathy in Japanese German Shepherd Dogs (GSD). Furthermore, the study proposed to analyse the clinical progression rate of the disease among dogs with a homozygous-mutant genotype.

I have read the authors’ reply and I am sorry to communicate that my recommendation is now to REJECT this paper.

The reason for my recommendation is the following: the authors are admittedly working on two different articles (this one and a future one “planning to report the relationship between the haplotype and clinical progression rate”) which are the same study tackled from two points of view. This survey on GSD could be easily incorporated in the multi-breed analysis on DM-related haplotype. Furthermore, there is a third article (“We are also surveying the Japanese BMD population […] we suppose that the survey will be finished in about a year”) which, even if on a different dog breed, could still be included in the same larger frame. Since results are on their way, authors are encouraged to report them in a single article giving an overall view on the subject, to avoid producing many redundant smaller papers.

An all-around article, including different dog breeds on the Japanese territory and the haplotype analysis, would be of utmost value for science regarding DM. I strongly encourage the authors to patiently wait for all the results before tailoring such a valuable article.

Reviewer 3 Report

The authors made improvements in their manuscript.

Still, some amendments are necessary.

Abstract

Line 37-42: please rephrase "This mutation is yet to be surveyed in the Japanese GSD population. Furthermore, the clinical progression rate among GSDs with the A/A DM-risk genotype is yet to be analyzed. Therefore, this study aimed to determine mutant allele frequency and to analyze the clinical progression rate in the Japanese GSD population. We surveyed 541 GSDs registered with the Japanese German Shepherd Dog Registration Society between 2000 and 2019. Genotyping was performed using real-time PCR with DNA extracted from the fur of each dog."

as similar as proposed here:

"This SOD1 mutation and the clinical progression rate of A/A risk genotypes in the Japanese GSD population have not been analysed before. Therefore, the aim of this study was to determine the frequency of the mutated allele and analyse the clinical progression rate in the Japanese GSD population. We studied 541 GSDs registered with the Japanese German Shepherd Dog Society between 2000 and 2019. Genotyping was performed using real-time PCR with DNA extracted from the hair roots of each dog."

Line 42-48: you may change this part as follows:

"The study revealed 330 G/G dogs (61%), 184 G/A dogs (34%) and 27 A/A dogs (5%), indicating a frequency of the mutant allele of 0.220, which are in Hardy-Weinberg equilibrium. We analysed the clinical signs in A/A dogs with an age limit of 10 years based on information obtained from the dogs' owners. Of the seven A/A dogs older than 10 years, owners reported DM-related clinical signs, indicating a clinical progression rate of 100%. These results, further genotyping and thorough clinical examinations of SOD1 A/A risk genotypes will help control and prevent DM in the Japanese GSD population."

It is absolutely necessary to say in the abstract how clinical progression was recorded.

Line 73-75: please rephrase. There are more data showing relationships among SOD1 genotypes and DM indicators in the paper cited and possibly in other reports. The authors should report these data in more detail. In addition, authors should provide definitions of progression rate used in previous reports. This is an important information for evaluation of the predictive value of the SOD1 test results.

Line 79-81: can be deleted because this is an opinion of the authors. No reference is given.

Line 91: The present study ...

LIne 123: (mentioned in the Introduction section): please delete.

Line 135-159:

It is confusing when authors are mixing up results with discussion. Thus, the authors should separate these parts. The usefulness of these statistical comparisons may be questioned as study designs and sampling strategies may not be comparable. 

Line 171-172: Two A/A dogs of unknown age had developed DM-related clinical signs.

Discussion

An important question is whether the dog owners got the SOD1 tests results prior the interview was made. This may influence the perception of the owner. You should discuss this issue.

You should also discuss whether breeding programs in GSD are regarding SOD1 genotypes and if there are mandatory regulations in respect to the SOD1 risk allele in place for breeding GSD in some countries or not.  

Why you can not show the development of the SOD1 genotype and allele frequencies by birth years. This would help to see if GSD breeders in Japan are concerned on the SOD1 risk allele and genotype.

Line 212-213: (a combination of five single nucleotide polymorphisms): this should be moved to Introduction.

Line 219-221: Such preventive measures may risk unnecessarily increasing the inbreeding coefficient of the PWC population and thus the risk of other genetic diseases.

Line 226: you should not say "diagnosis of DM" because you used owners' reports on DM-related clinical signs. It would be useful to discuss the possible differential diagnoses which have to be regarded and whether this issue was addressed during the interviews.

Conclusions

The authors have to say how the results of the SOD1 genotypes are implemented in the GSD breeding program in Japan and which breeding progress has to be expected. Are there specific regulations for the licensing of sires and dams. Please work out in more detail.
